# Thermodynamical Analysis of the Formation of *α*-Si Ring Structures on Silicon Surface

**DOI:** 10.3390/ma16062205

**Published:** 2023-03-09

**Authors:** Vygandas Jarutis, Domas Paipulas, Vytautas Jukna

**Affiliations:** Laser Research Center, Vilnius University, Sauletekio Avenue 10, LT-10223 Vilnius, Lithuania

**Keywords:** ultrafast lasers, laser micromachining, silicon ablation

## Abstract

Superficial modifications on silicon wafers produced by single-shot focused femtosecond laser irradiation having a 1030 nm wavelength and 300 fs pulse duration were experimentally and theoretically analyzed. The laser fluence window when the amorphous silicon phase develops, resulting in a ring-like modification shape, was experimentally estimated to be between 0.26 J/cm${}^{2}$ and 0.40 J/cm${}^{2}$ and was independent of the silicon dopant type and laser focusing conditions; however, the window was narrower when compared to results reported for shorter pulse durations. In addition, we present a simplified numerical model that can explain and predict the formation of these patterns based on the caloric coefficients of silicon and the energy distribution of the deposited material.

## 1. Introduction

Direct laser writing (DLW) and surface patterning with ultrafast laser pulses remain one of the most versatile methods for tailoring material properties [1,2]. A wide range of light–matter interaction phenomena could lead to different technological applications such as precise laser ablation, enabling laser micromachining with minimal thermal impact [3], and the self-organized generation of nanostructures on the surface [4] helps to achieve unique surface properties while a selective modification of materials’ optical properties can open new ways for integrated photonics and similar applications [5]. The universality of light absorption mechanisms for ultrafast laser pulses makes the DLW method applicable for virtually all materials: dielectrics, metals, and semiconductors; however, the reaction of each material to the absorbed radiation may vary.

For example, in semiconductors, such as crystalline silicon (*c*-Si), it is very common to observe amorphized regions (α-Si) at the boundaries of laser-irradiated areas. It is generally accepted that amorphization in silicon happens when the molten silicon layer cools down at a rate such that the resolidification front moves faster than the critical speed, which is in the range of 1–25 m/s [6]. At this cooling speed, the nucleation process cannot develop, resulting in amorphous silicon. The high thermal conductivity of silicon can efficiently cool down the solid–liquid interface and reach ultracritical speed in thin liquid layers. As the refractive index of (α-Si) is slightly higher than that of *c*-Si [7], resulting in a higher reflectivity under visible light, this region can be clearly distinguished using an optical microscope. The typical ultrafast laser fluence window for amorphization can be found in the literature and is in the range of 0.26 J/cm${}^{2}$–0.52 J/cm${}^{2}$ [8]. At higher fluences, the amorphized regions abruptly disappear, indicating silicon recrystallization. Therefore, a bright ring-shaped pattern can be observed if the laser’s fluence is higher than the indicated window. Such rings were first observed by Bloembergen and Liu in the 1980s using the first picosecond lasers, while the α-Si phase was confirmed using electron diffraction patterns [9]. Utilizing the sharp visibility of the α-Si phase, Liu suggested a simple method for measuring the focused laser waist size, assuming that threshold values were constant and depended only on the fluence [10]. In later studies, it was shown that more complex structures could form on silicon after local melting induced by a single ultrafast laser pulse, including amorphization, recrystallization, native oxide removal or melting, oxidation, and ablation [8,11], each of which has its own threshold fluence.

The selective superficial amorphization of silicon in tandem with ultrafast DLW has several emerging applications. α-Si has a lower selectivity for wet chemical etching that can be utilized for maskless lithography [12], and DLW flexibility and selective surface patterning [13] can be used for custom MEMS devices. In addition, laser patterning can generate features with subwavelength resolution, as demonstrated by the production of highly periodic LIPPS with alternating α-Si and *c*-Si regions [14,15,16,17]. α-Si surrounded by *c*-Si can support waveguiding [18] and the research on the DLW integration of such devices was carried out in several studies [19]; however, the successful integration has yet to be demonstrated—the laser-produced α-Si layer is too thin and prevents waveguiding for practical wavelengths. Using fluences well over the ablation threshold [20] or selecting the laser wavelength inside the silicon transparency window [19] still produces an amorphous layer of tens of nanometers in thickness.

The extensive real-time dynamic studies of semiconductor behavior under a femtosecond laser pulse were carried out by the Siegel and Bonse groups, which helped to establish time scales as well as experimentally characterize the damage/modification morphology [17,21,22]. However, there is still a lack of theoretical analysis that could explain the amorphization effects from light–matter interaction effects and the thermodynamic properties of silicon. In this study, we investigated the formation of amorphous ring structures on the silicon surface methodically and examined the ring diameter dependence on irradiation conditions and the silicon material itself (different doping concentrations and crystal orientations). The analysis of the experimental results leads to a theoretical investigation. The derived simple numerical simulations give a very good match to the experimental data when laser-deposited energy is connected to the caloric coefficients.

## 2. Experiment

An experiment was conducted to investigate the formation of ring-shaped structures on the silicon surface after single-shot laser irradiation. An Yb:KGW laser Pharos (Light Conversion Ltd., Vilnius, Lithuania) with a beam quality of $M^{2}$ equal to 1.1 that operated at a 1030 nm wavelength and produced 300 fs duration pulses was used as a light source. A silicon sample was placed on the XY translation stage ANT 180 (Aerotech, Pittsburgh, PA, USA), while a laser beam was focused with a 100 mm lens mounted on the Z axis translation stage ANT 130 (Aerotech, Pittsburgh, PA, USA). The beam waist at the lens focus was 15.4 μm (measured by Liu’s method [10]). Several different beam diameters were also tested in the experiment, which were produced by shifting the sample slightly above the geometrical focus as shown in Figure 1. In this way, we were able to analyze the ring formation dependence on the beam diameter. The silicon surface was patterned in an X–Y grid with different pulse energies, which were attenuated from 0.2 μJ (no visible damage was observed on a sample at such energy) up to 8 μJ. All experimental procedures were automated using the laser microfabrication software application SCA (Workshop of Photonics, Vilnius, Lithuania)). Samples were investigated using an optical microscope profilometer, PLμ2300 (Sensofar Ltd., Barcelona, Spain) with an EPI 100X, NA = 0.9 objective, and no morphological surface change was observed till the damage of the material. Therefore, an optical microscope was used at the standard regime, and images were processed to extract the inner and outer ring diameters of amorphized regions. Scanning electron imaging was also performed; however, due to the poor contrast of the images of the ring structures, it was not investigated. Selected samples were imaged with an Olympus BX51 microscope using an oil immersion objective with NA 1.25. These are depicted in Figure 2.

In total, six polished silicon samples with different dopant concentrations and crystal orientations were used in the experiment (see Table 1). A typical result of a single laser pulse effect on the silicon surface is shown in Figure 2. When laser fluence is relatively modest, the irradiated spot appears brighter in microscope images because the amorphized silicon region, which has a higher refractive index, reflects more light. At a certain fluence, however, the reflectance reduces sharply, resulting in a well-defined ring-shaped pattern that is clearly seen in the pictures. With increasing fluence, the ring diameter expands while the ring width decreases, indicating the existence of a specific fluence window for silicon amorphization. The rings are only slightly irregular due to the quality of the laser beam, which was measured to be $M^{2}=1.1$. The ring’s inner and outer diameters were measured by fitting two coaxial circles that best fit the data (see Figure 2 bottom right image). We did not identify any variations in ring formation thresholds between different dopant levels in silicon samples. A slight reduction of damage threshold for doped silicon might be visible in Figure 2 when the highest fluence was used. There was a striking difference in crystal orientation: amorphized rings were only distinguishable in the 〈111〉 crystal orientation cut wafers, whereas the 〈100〉 samples were scarcely discernible—only with digitally enhanced images was the modified region evident, as seen in the bottom section of Figure 2.

Despite the difference in visibility, the modified region size for 〈111〉 samples and 〈100〉 samples under the same focusing condition was nearly identical, indicating the same modification threshold. In this paper’s Appendix A, we provide raw optical microscope images comparing the modifications on all samples under identical focusing conditions (see Animation S1), as well as confocal microscope images made with a monochromatic (460 nm) LED illumination with enhanced contrast (see Animation S2). The later regime helped to identify and measure the boundaries of the inner and outer modification regions with greater clarity; therefore, these data were used in the analysis.

It was reported that high-contrast amorphized regions in 〈100〉 silicon developed only after multipulse irradiation [13]; nevertheless, Florian et al. showed that clear rings could also be produced with single-pulse irradiation [11]; however, in that study, the laser source had a pulse duration of up to an order of magnitude shorter than used in our study, demonstrating that pulse duration may be critical even at the ultrafast time scales.

Due to the greater visibility of the rings, the 〈111〉 silicon wafers were extensively investigated and used for systematic measurements. The Si 〈100〉 orientation laser-generated amorphization rings were faint, therefore difficult to accurately measure. Only the outer ring radius was measured for the smallest beam diameter and is depicted by squares in Figure 3b,e,h. The Si 〈100〉 and Si 〈111〉 orientations’ outer ring radius were similar, the Si 〈100〉 results being slightly scattered due to the poor contrast of the rings, leading to errors in measurement.


For each type of silicon doping and beam diameter, we analyzed the dependence of the inner and outer ring radii on the beam’s maximum fluence. The results for silicon without impurities are shown in Figure 3a–c, for n-type silicon in Figure 3d–f and for (p-type) silicon in Figure 3g–i. Figure 3a,d,g show that for all types of silicon and all focusing conditions, there was a fluence threshold below which the inner ring was not observed. The same conclusion followed for the outer ring (see Figure 3b,e,h. To determine the threshold fluence values $F_{\mathrm{thr}.}^{\left(i,o\right)}$ for the inner and outer radii, we used the empirical formula
(1)$$F_{\mathrm{thr}.}^{\left(i,o\right)}=Fe^{-r_{i,o}^{2}/w^{2}}\;,$$
where *w* is the laser beam radius on the surface of a crystal, $r_{i}$ is the inner radius of the ring, $r_{o}$ is the outer radius of the ring, and *F* is the fluence of a laser pulse at the beam center. The solid lines in Figure 3 depict the fit of the empiric equation which demonstrated a very good match with experiments when the fluence threshold value for the outer and inner ring radii was 0.26 and 0.4 J/cm${}^{2}$, respectively. Equation (Equation 1) also suggests that normalized values of the radii depend only on the pulse fluence and this is exactly what was observed in the experiment. The ring radii dependence on fluence for all the beam diameters are depicted in Figure 3c,f,i, and all of them fit perfectly.

## 3. Theory

The formation of the ring-shaped structure on the crystal surface is directly related to the amount of absorbed energy of the incident pulse and how this energy is distributed inside the crystal. Our hypothesis is that the absorbed energy density value $\rho_{E}\left(r,z\right)$, where *z* is the distance from the crystal surface and *r* is the distance from the laser beam center (cylindrical symmetry is assumed), determines the outer and inner ring radii essentially. As a result, the energy density must have two critical values, $\rho_{E1}$ and $\rho_{E2}$, which correspond to the outer and inner radii of the ring on the surface (z=0):(2)$$\rho_{E}\left(r_{o},0\right)=\rho_{E1}\;,\;\rho_{E}\left(r_{i},0\right)=\rho_{E2}\;.$$ Here, $\rho_{E1}$ is the energy density needed to raise the local temperature of the silicon from room temperature up to the melting temperature $T_{m}$, and $\rho_{E2}=\rho_{E1}+\rho_{EL}$, where $\rho_{EL}$ is the latent energy density needed to transform solid silicon to the liquid phase. It is well-known from elementary thermodynamics that when a quantity of heat dQ is applied to a body of mass *m*, its temperature changes according to the law mCdT=dQ, where *C* is the specific heat of that material. Taking into account that m=ρV, we rewrite our equation in the form $\rho CdT=dQ/V\equiv d\rho_{E}$. From this expression, we can find the critical energy density corresponding to the melting temperature
(3)$$\rho_{E1}=\rho \int_{T_{0}}^{T_{m}}C\left(T\right)dT\;,$$
where ρ is the density, and $T_{0}$ is the initial temperature of the silicon. Given that the isobaric and isochoric specific heats of solids differ only slightly, we assume that $C=c_{p}\left(T\right)/M$, where *M* is the molar mass and $c_{p}\left(T\right)$ is the molar isobaric heat of silicon (see Table 2). Evaluating the integral, we find $\rho_{E1}=2.987$ nJ/μm${}^{3}$. If the silicon’s molar latent heat is denoted as *L* (J/mol), then the corresponding energy density is $\rho_{EL}=\rho \;L/M=4.171$ nJ/μm${}^{3}$, and consequently, $\rho_{E2}=7.158$ nJ/μm${}^{3}$.

In order to estimate the amount of absorbed energy density in the crystal, we considered a simplified model of the interaction between laser radiation and matter [25]. Given the short pulse duration, we accounted only for single and two-photon absorption, free carrier absorption (FCA), carrier relaxation, and Auger recombination but ignored heat transport. The characteristic heat transport distance per pulse duration can be estimated using the expression $l={\left(K\tau_{1/2}/\left(\rho C\right)\right)}^{1/2}$ which is derived from the fundamental solution of heat diffusion equation. For silicon, the typical value of the thermal conductivity is K=1.3 W/(cm·K) and therefore, l≈4 nm which is an order $10^{4}$ times smaller than the laser beam diameter on the crystal surface or pulse propagation distance inside the crystal. In such circumstances, the effect of heat diffusion is negligible and, to a first approximation, can be ignored. The system of equations describing the silicon–radiation interaction is
(4)$$\begin{array}{cc} \frac{\partial I}{\partial z} & =-\alpha \left(N\right)I-\beta I^{2}\;, \end{array}$$
(5)$$\begin{array}{cc} \frac{\partial N}{\partial t} & =\frac{\alpha_{\mathrm{SPA}}}{\hbar \omega }\;I+\frac{\beta }{2\hbar \omega }\;I^{2}-\frac{N}{\tau_{R}}-\gamma N^{3}\;. \end{array}$$
Here, I(r,z,t) denotes the intensity of the laser pulse, N(r,z,t) is the free electron concentration, β is the two-photon absorption coefficient, ℏω is the photon energy, $\tau_{R}$ is the relaxation time constant of the electron concentration, and γ is the coefficient for Auger recombination. Absorption coefficient α accounts for the intrinsic silicon absorption and the absorption due to free electrons:(6)α(N)=2ωcIm[n˜(N)](7)$$\begin{array}{cc} \tilde{n}\left(N\right) & =\sqrt{{\tilde{\varepsilon }}_{i}-\frac{Ne^{2}}{\varepsilon_{0}\omega^{2}m_{e,\mathrm{opt}.}^{*}}\;\frac{1}{\left(1+\frac{i}{\omega \tau_{c}}\right)}}\;, \end{array}$$
where $m_{e,\mathrm{opt}.}^{*}$ is the joint optical effective mass of an electron in a crystal, $1/m_{e,\mathrm{opt}.}^{*}=1/m_{e,c}^{*}+1/m_{h,c}^{*}$, $\tau_{c}=1/\nu_{c}$ is the collision period of electrons and holes with the lattice. For silicon, the exact value of the collision frequency $\nu_{c}$ is not known, but it could be estimated from expression $\nu_{c}=AT$, where proportional constant $A=3.9\times 10^{11}$ s${}^{-1}$K${}^{-1}$ [25]. From this expression, it follows that the collision period varies from $\tau_{c}=33$ fs for 300 K down to 8.3 fs for $T=T_{m}$. In our model, we used $\tau_{c}$ as a free parameter. The coefficient $\alpha_{\mathrm{SPA}}=\alpha \left(0\right)$ accounts only for single-photon absorption due to bound electrons. The boundary conditions are
(8)$$\begin{array}{cc} N(r,z,-\infty ) & =N_{0}\;,\;z⩾0\;, \end{array}$$
(9)$$\begin{array}{cc} I(r,0,t) & =\left(1-R\right)I_{0}\;\exp\left(-2\;\frac{t^{2}}{\tau_{p}^{2}}-2\;\frac{r^{2}}{w^{2}}\right)\;, \end{array}$$
where $N_{0}=10^{10}$ cm${}^{-3}$ is the free electron concentration at room temperature, *w* is the beam radius on the crystal surface, $\tau_{p}=\tau_{1/2}/\sqrt{2\ln2}$, $I_{0}=\left(\sqrt{2/\pi }/\tau_{p}\right)F_{0}$, $F_{0}$ is the laser pulse fluence, and *R* is the reflectivity under normal incidence
(10)$$R\left(N\right)=\frac{|\tilde{n}{\left(N\right)-1|}^{2}}{|\tilde{n}{\left(N\right)+1|}^{2}}\;.$$
With the solutions of Equations (Equation 4) and (Equation 5), we can calculate the energy density absorbed in silicon using the expression
(11)$$\rho_{E}\left(r,z\right)=-\frac{\partial F}{\partial z}=\int_{-\infty }^{\infty }\left(\alpha I+\beta I^{2}\right)dt\;.$$

During the simulation, we varied three parameters: the electron and lattice collision period $\tau_{c}$, the electron concentration relaxation time $\tau_{R}$, and the two-photon absorption coefficient β. The best agreement with the experiment was obtained when $\tau_{c}=64.6$ fs, $\tau_{R}=161$ fs, and β=18.0 cm/GW. In particular, on the crystal surface for this set of parameters, we obtained early calculated absorbed energy densities $\rho_{E1}$ and $\rho_{E2}$ for fluences $F_{1}=0.260$ J/cm${}^{2}$ and $F_{2}=0.408$ J/cm${}^{2}$, respectively, which were in very good agreement with the experimental values of threshold fluences (see Figure 4a–c). An interesting feature of the model was that it predicted a relatively sharp outer edge of the ring-shaped structure (see Figure 4d). It was directly related to critical electron concentration $N_{c}$ which could be determined from equation Re(n˜2)=0. In our case, $N_{c}=0.0409\times N_{at}$, where $N_{at}=4.99\times 10^{22}$ cm${}^{-3}$ was the atom concentration. Our simulations showed that as soon as the electron concentration approached the critical value $N_{c}$ (Figure 4g), and there was enough pulse energy, the effective absorption coefficient α started to grow rapidly (Figure 4e) and the absorption rate increased significantly (Figure 4f).


The dependence of the absorbed energy density on the crystal depth over the interaction time (about 1 ps) is shown in Figure 5. The process of crystal amorphization ($\rho_{E}>\rho_{E1}$) began when the energy fluence exceeded the value $F_{1}=0.26$ J/cm${}^{2}$. It is important to note that the amorphization depth found from the graph represented only the lower value of the real depth. For example, the model predicted that the amorphization depth was 23 nm for the fluence F=0.32 J/cm${}^{2}$ (see the line marked by the ▽ symbol), whereas the experiment showed it to be around 45 nm [19]. One reason for this discrepancy is the fact that we did not take into account the heat transport after the interaction. Knowing that the laser beam size on the surface is of the order of μm, and the absorbed energy depth is of the order of nm, due to the gradient of heat, the heat transport is mostly in the depth direction. The generation of shock waves might also help to deepen the amorphization layer [26]. Another and potentially more important reason is that the plasma gets very reflective and changes the group velocity of the pulse when it reaches close to the critical plasma density, which also may change the depth of the amorphization.

Despite a lot of postpulse effects emitting drawbacks in our numerical analysis, we showed that this simplified numerical model which did not demand a high computation power worked surprisingly well on the analysis of the surface modification of silicon. For example, we found experimentally that the surface ablation threshold for undoped silicon was about 0.62 J/cm${}^{2}$. For such fluence, the absorbed energy density on the surface was very close to the limiting value $\rho_{E3}=11.334$ nJ/μm${}^{3}$ required to reach the boiling point of silicon (see Figure 5, line marked by ⋄ symbol).

The simplified numerical model suggested that a ring-shaped structure on the silicon surface was formed due to a change in the aggregate state of the material. From a thermodynamic point, it is safe to say that when $r>r_{o}$, the crystal surface remains unaffected. In the region $r_{o}<r<r_{i}$, one has a transient solid–liquid state where $T=T_{m}$. In that region, bonds between atoms begin to break and the formation of randomly oriented and partially oxidized microcrystals is possible. The random orientation of microcrystals can explain the strong light scattering from the surface. In the region $r<r_{i}$, all silicon on the surface should be liquid for some time. It is very difficult to expect that silicon in this region can recrystallize into the former lattice without any consequences. The formation of the oxide layer or modification of the native oxide layer is not excluded, and it can form on the silicon surface, especially when it is heated and in a liquid state [11]. However, other experiments in inert gases or liquids also show the same ring formation [27]. When laser energy absorption is high enough, i.e., $\rho_{E}>11.3$ nJ/μm${}^{3}$, the boiling of silicon can be reached, and this can be associated with the ablation threshold of the material.

## 4. Summary and Conclusions

We showed the formation of rings on the surface of silicon with various beam diameters and provided a simplified numerical tool that accurately predicted the ring formation when the latent heat was included. We experimentally demonstrated that the laser fluence threshold of the outer ring and inner ring does not depend on the beam diameter, material doping, or crystal orientation, when it is irradiated with a 1030 nm wavelength and 300 fs duration laser pulses. Surprisingly, the numerically estimated depth of the amorphous layer was similar to what other research groups measured. This simplified numerical tool can be used to further analyze the amorphization of the silicon surface with altered beam shapes and/or pulse duration. It was shown that the depth of the amorphization was clearly dependent on the laser energy absorption and heat; we think the use of the laser burst mode (a couple of pulses that are separated in time with delays much shorter than the repetition rate of the laser) could increase the depth of the heat transfer and increase the depth of the amorphous layer.

## Figures and Tables

**Figure 1 materials-16-02205-f001:**
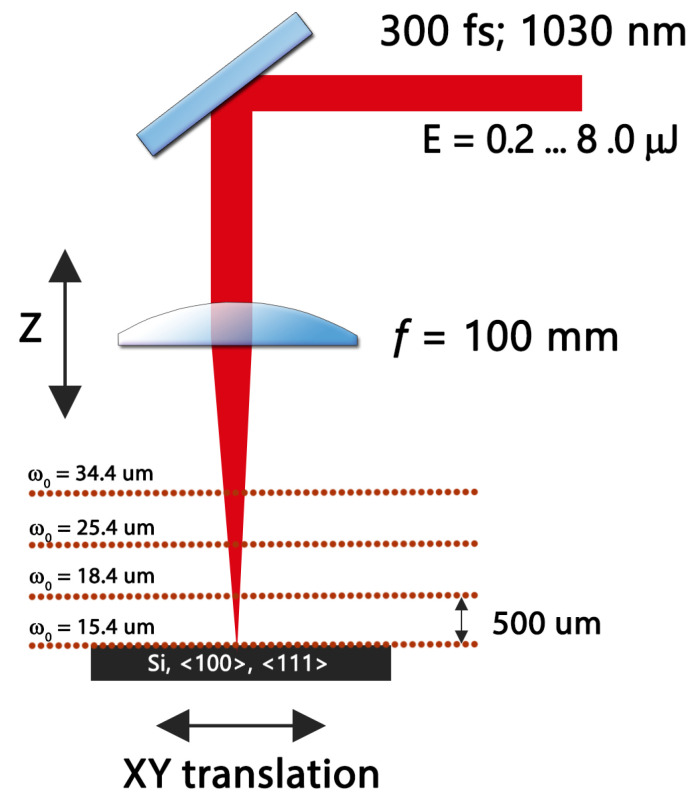
Experimental setup of single-pulse femtosecond laser processing of silicon surface.

**Figure 2 materials-16-02205-f002:**
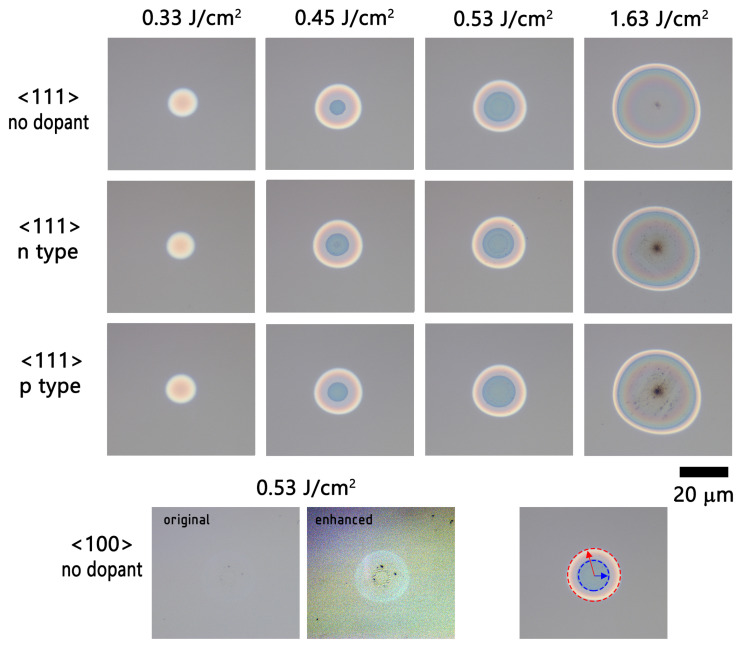
Optical microscope images of silicon crystal following single-shot laser irradiation at various laser fluences. The first three rows depict images for pristine, n-, and p-type 〈111〉 crystal while the bottom row shows a typical damage image for 〈100〉 crystal, with both the original and contrast-enhanced images provided for clarity. The boundaries of the outer and inner rings are separately marked at the bottom right image.

**Figure 3 materials-16-02205-f003:**
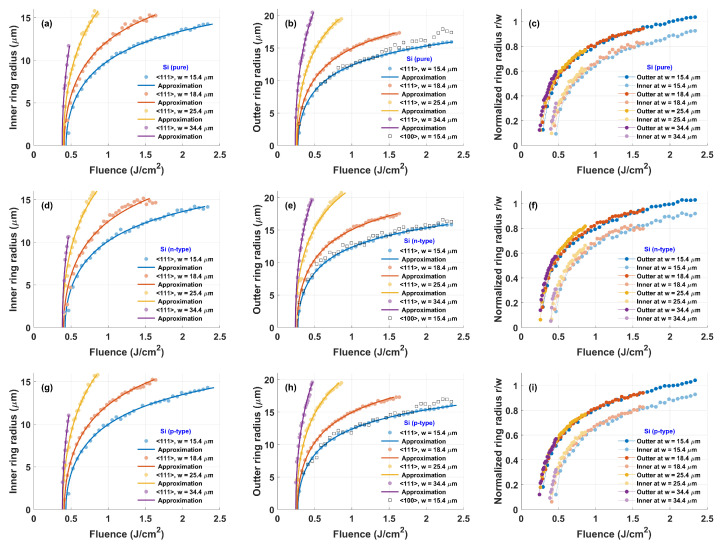
The inner (**a**,**d**,**g**), outer (**b**,**e**,**h**), and normalized (**c**,**f**,**i**) radii dependence on the fluence of laser pulse and beam focusing conditions. Circles depict the Si 〈111〉 crystal orientation while squares depict the Si 〈100〉 crystal orientation. The points correspond to the experimental measurements and the solid line to the approximation $F_{\mathrm{thr}.}^{\left(i,o\right)}=F\exp\left(-r_{i,o}^{2}/w^{2}\right)$. Silicon type: (**a**–**c**) no dopant, (**d**–**f**) n-type, (**g**–**i**) p-type.

**Figure 4 materials-16-02205-f004:**
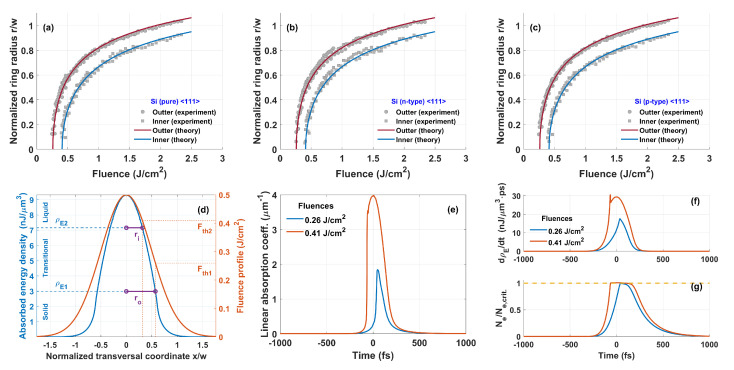
The inner and outer radii dependence on the fluence for (**a**) pure, (**b**) n-type, and (**c**) p-type silicon. Points correspond to the experimental measurements. The solid lines calculated numerically for $\tau_{c}=64.6$ fs, $\tau_{R}=161$ fs, and β=18.0 cm/GW. (**d**) Absorbed energy density on the surface across the beam. (**e**) Effective absorption coefficient α, (**f**) energy density absorption rate, and (**g**) electron concentration on the crystal surface for two limiting fluence values $F_{1}$ and $F_{2}$.

**Figure 5 materials-16-02205-f005:**
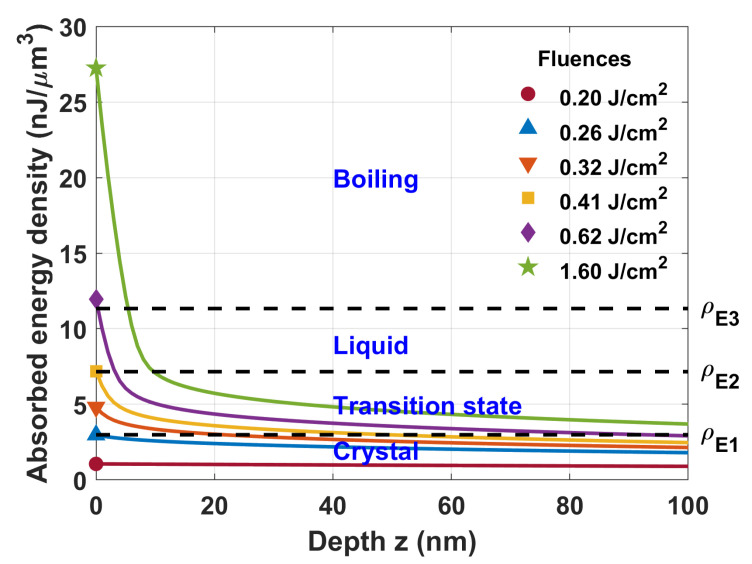
Calculated absorbed energy density as a function of depth for different fluence values. Model parameters: $\tau_{c}=64.6$ fs, $\tau_{R}=161$ fs, and β=18.0 cm/GW. Limiting energy densities: $\rho_{E1}=2.987$ nJ/μm${}^{3}$ (requires to reach the melting point), $\rho_{E2}=7.158$ nJ/μm${}^{3}$ (requires to transform solid silicon into liquid phase), $\rho_{E3}=11.334$μm${}^{3}$ (requires to reach the boiling point).

**Table 1 materials-16-02205-t001:** Silicon samples.

Part No. (Sigma-Aldrich, Burlington, VT, USA)	Crystal Orientation	Type	Resistivity (Ωcm)
647101	〈111〉	No dopant	100–3000
647705	〈111〉	p-type (boron as dopant)	$10^{-3}$–40
647799	〈111〉	n-type (phosphorus as dopant)	$10^{-3}$–40
646687	〈100〉	No dopant	100–3000
647675	〈100〉	p-type (boron as dopant)	$10^{-3}$–40
647780	〈100〉	n-type (phosphorus as dopant)	$10^{-3}$–40

**Table 2 materials-16-02205-t002:** Modeling parameters. The coefficients for the isobaric specific heat equation are A=22.81719, B=3.89951, C=−0.082885, D=0.042111, E=−0.35406, and t=T/1000, *T* in K.

Quantity	Symbol	Value
Silicon parameters		
Density	ρ	2.33 g/cm${}^{3}$
Molar mass	*M*	28.0855 g/mol
Initial temperature	$T_{0}$	300 K
Melting temperature	$T_{m}$	1687 K
Molar latent heat [23]	*L*	50,208 J/mol
Molar isobaric specific heat [24]	$c_{p}\left(T⩽T_{m}\right)$	$A+Bt+Ct^{2}+Dt^{3}+E/t^{2}$, J/(mol·K)
	$c_{p}\left(T>T_{m}\right)$	27.196, J/(mol·K)
Intrinsic dielectric constant	$\tilde{\varepsilon }$ at $\lambda_{0}$	$12.7104+1.71169\times 10^{-3}\;i$
Electron conductivity effective mass	$m_{e,c}^{*}$	0.26 $m_{e}$
Hole conductivity effective mass	$m_{h,c}^{*}$	0.37 $m_{e}$
Auger recombination coefficient	γ	$3.8\times 10^{-31}$ cm${}^{6}$/s
Laser parameters		
Wavelength	$\lambda_{0}$	1.03μm
Pulse duration at FWHM	$\tau_{1/2}$	300 fs

## Data Availability

Any further details relevant to this study may be obtained from the authors upon a reasonable request.

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
