# Peer review of "Thermodynamical Analysis of the Formation of α-Si Ring Structures on Silicon Surface"

_materials, 2023, doi:10.3390/ma16062205_

Round 1

Reviewer 1 Report

In the article of Jarutis et at. is studied the modification of silicon by single-shot femtosecond lasers, and more in particular the origin of the formation of an amorphous ring when irradiating with pulses with special Gaussian distribution. The authors performed irradiations with a laser emitting 300-fs pulses at 1030-nm wavelength. The authors determine the fluence window (lower limit and upper limit) in which the amorphous phase appears for different kind of silicon sample (different doping: n-type, p-type and intrinsic; and different orientations: 100 and 111). From the experimental point of view, the determination of those values seems accurate and precise since it was demonstrated the consistency by using different focusing conditions.

Additionally, and being the aspect that presents the major novelty of the article, authors performed numerical simulations accounting the thermodynamical processes leading to the formation of the amorphous ring. As mentioned, this topic is of importance and it is really worthy to be studied. However, precisely on this modelling, I consider the article presents two major drawbacks:

1.       The physical criteria limits chosen for the formation of amorphous phase.

While agreeing on the criteria for the lower limit (heating of the solid up to the melting temperature), I consider the upper limit criteria wrong. The authors associate this level to the energy needed to melt from solid to liquid, accounting apart from the heating up to the melting point, the latent heat to melt the material. However, this criterion excludes that the liquid silicon reaches temperatures above the melting temperature in contradiction with the bibliography on the topic. The article “Garcia-Lechuga, M. et al. ACS Photonics 3, 1961–1967 (2016)”, where time-resolved reflectivity measurements are performed, reports the observation of overheated liquid silicon reaching temperatures above the melting point and below the boiling temperature on regions where finally the material re-solidifies in amorphous phase. Therefore, the heat absorption criterion on this article for the upper limit is clearly underestimated. In a simplify case not accounting the complex dynamics of re-solidification, a more logical upper limit could be the heat needed to reach the boiling temperature.

2.       The reported maximum liquid thickness

The first mentioned drawback has consequences on the following reported values. In particular, when looking at figure 5, two surprising results can be seen. First, the maximum layer thickness of the liquid layer obtained is 5 nm. This value is well bellow the maximum amorphization thicknesses reported on the bibliography (cited on the article) when irradiating silicon <111> with similar wavelengths: 65 nm in ref [18] and 60 nm in ref [11]. Obviously, for generating this phase-change is needed at least the same thickness of molten material. Secondly, the reported maximum thickness of liquid silicon is reached at a fluence value (1.6 J/cm2) that is clearly above the fluence threshold of ablation which is on contradiction with the amorphization process.

Thus, the fact that the main novelty of the article is the thermodynamical modelling and considering the two drawbacks, I consider that the article should not be consider to be published in Materials.

Reviewer 2 Report

The authors studied the thermodynamic behavior of the formation of α-Si ring structures on silicon wafer surface. The research is very meaningful for ultrafast laser processing. However, this article is slightly inadequate in terms of experiments as a result of insufficient supports for theory. The following suggestions for reference.

1. Femtosecond laser processing of silicon wafers has been studied for many years. It is suggested to supply the development trend of relevant researches in the formation of Figures (in the part of Introduction) to highlight the innovation of this paper better.

2. The ablation thresholds of silicon wafers with different crystal orientations and doping types are also different. Is it considered in this paper?

3. Please add at least one enlarged SEM image and mark the measured areas in the image to distinguish the size of the ring.

4. Does the amorphization of silicon wafers have obvious directivity in the processing area? Is it evenly distributed along the radial and axial directions of the wafer? Are the amorphization affected by orientation and doping type of silicon wafers? Please add TEM photos to prove this phenomenon.

Reviewer 3 Report

In this paper, the authors analyzed the formation of ring-structured amorphous silicon layer under laser writing. While the results are interesting, I am afraid that this manuscript cannot be accepted in the current form and major revisions must be done before this manuscript can be published on any research journal.

(1)   There are plenty of grammatic errors and typos in this manuscript. There are many sentences embedding in each other that once cannot find the proper subject or predicate. This makes the manuscript much less reader-friendly.

(2)   The authors need to highlight the novelty and significance of their work properly in the manuscript, esp in the abstract and introduction. I failed to find any new physical insight or experimental understandings that distinguish from previous works.

(3)   Not all technical details are given in the manuscript.

(4)   In addition to the peculiar halos under optical scope, I am extremely curious about the morphology of the treated region of silicon wafer.

(5)   Fig 3 lacks proper description in the manuscript. There are 9 panels in Figure 3, but there is only one sentence describing them? What is the difference between all 9 panels?

(6)   The figure quality need significant improvements. The authors need to increase the font size and move the unit of axis title into brackets.  

Round 2

Reviewer 1 Report

Comments on the file attached.

Reviewer 2 Report

It is suggested to publish this article.

Author Response

We thank the reviewer for accepting our manuscript to be published.